# R-VLA: Residual Off-Policy Adaptation of a Frozen Vision-Language-Action Policy

Srihith Mudakala and Arnav Patil

*Abstract*—Adapting a Vision-Language-Action model (VLA) to a deployment task by fine-tuning its backbone can require substantial memory, differentiable model access, and expert demonstrations. We study a complementary setting in which the VLA remains frozen and a small residual policy learns bounded action corrections with off-policy Soft Actor-Critic (SAC). The residual actor observes robot proprioception and the VLA action, while a trainingonly asymmetric critic additionally observes privileged simulator state. Caching the deterministic VLA action for the current and next replay states eliminates backbone calls during gradient updates and yields a measured $25\times$ wall-clock reduction relative to recomputation on our hardware. Rewards are supplied through a common interface, enabling generated, hand-written dense, and sparse reward conditions. After correcting two evaluation-protocol bugs, we screen the frozen policy on 16 LIBERO tasks and train residual policies on three tasks for 10,000 environment steps with three seeds. Within this budget, trained policies remain below separately estimated frozen-policy anchors. A reward audit finds that generated-reward return can rise while task success remains low, with return–success correlations from $-0.21$ to $0.66$ and prolonged object-contact behavior. These preliminary, simulation-only results do not demonstrate successful policy correction; they establish an architecture, an anchor-first protocol, and diagnostics for testing residual VLA adaptation.

## I. INTRODUCTION

Vision-Language-Action models have emerged as a compelling paradigm for generalist robot manipulation: a single pretrained model, conditioned on a natural-language instruction and a camera image, directly predicts robot actions [6, 5]. Models trained on large multi-task datasets such as Open-X-Embodiment exhibit impressive zero-shot transfer, but their performance degrades substantially when deployment conditions differ from training in object geometry, lighting, or task specification phrasing [6].

**Motivation: adaptation without backbone training.** A VLA that underperforms on a target task can be adapted by full or parameter-efficient fine-tuning, but both require differentiable backbone access and task-specific data; full fine-tuning additionally incurs the optimizer and activation memory of a large multimodal model. We instead ask how much adaptation is possible when the pretrained policy is held fixed and only a small controller is trained around it. Simulation supplies a task reward and privileged state during training, while the deployed controller must act from signals available on the robot.

We freeze the VLA and train a *residual correction policy* whose bounded output is added to the VLA action. The actor is restricted to proprioception and the VLA action, so it does not depend on simulator-only observations. Its mean head is zero-initialized: before gradient updates, deterministic mean-action evaluation exactly reproduces the frozen policy. Only the residual actor and critics receive gradients.

**Problem setting.** Training occurs in simulation, where programmatic rewards and privileged object state are available. To study reward design independently of the RL implementation, rewards enter through a fixed `ExternalReward` interface. This supports LLM-generated functions following the GRACE methodology [9], hand-written dense rewards, and the environment's sparse success signal without changing SAC. Privileged pose and contact signals may enter the reward and critic during training but never the actor.

We use **Soft Actor-Critic (SAC)** [3] for off-policy replay. Because the frozen VLA is deterministic, its actions for the current and next states can be stored with each transition; replay updates then require no VLA forward passes. This is a compute and wall-clock optimization rather than evidence of improved environment-sample efficiency. An asymmetric critic uses privileged simulation state during training, while the actor retains the deployable observation boundary.

This paper contributes: (1) a residual SAC architecture over a frozen VLA, with deployable actor inputs and bounded corrections; (2) cached-prior replay and an asymmetric actor-critic; (3) a common interface for generated, hand-written, and sparse rewards; and (4) an anchor-first LIBERO protocol with return–success and behavior diagnostics. We also document two corrections—a gripper-sign inversion and an identical-init-state evaluation bug—that invalidate our earlier workshop-submission numbers.

We position this as an early-stage systems study, not a state-of-the-art or completed sim-to-real result. Section IV reports the quantitative evaluation and Section VI states the evidence boundary.

## II. RELATED WORK

### A. VLA Fine-Tuning Approaches

Full fine-tuning of VLAs on target-domain datasets yields the strongest task-specific performance but risks degrading cross-task generalization and demands GPU memory that may be unavailable at the edge [6]. LoRA and other parameter-efficient methods reduce the number of updated parameters but still require backward passes through the backbone and target-domain expert demonstrations. Our approach requires neither backbone gradient access nor expert demonstrations.

## B. Residual Policy Learning

Residual RL adds a learned correction to a fixed nominal controller, allowing standard RL to focus on a smaller correction signal rather than the full action space [11, 4]. Applying this to large pretrained VLAs as the base policy is less studied. Our work extends the residual RL paradigm to the VLA setting, with the frozen-backbone property enabling the cached-prior replay optimization described in Section III.

## C. LLM-Based Reward Generation

Ma et al. [8] proposed Eureka, which uses an LLM to iteratively generate and refine reward code guided by scalar environment feedback. Xie et al. [12] introduced Text2Reward, which produces dense, interpretable reward functions from task descriptions. Sapora et al. [9] introduced GRACE, a language model framework for generating explainable reward functions grounded in inverse reinforcement learning, targeting precise manipulation reward specification. All three share a common interface—the LLM returns executable Python producing a scalar reward and a diagnostic dictionary—which our `ExternalReward` wrapper adopts directly, making the reward component interchangeable.

## D. Off-Policy Actor-Critic Methods

SAC [3] combines off-policy replay, twin-Q critics with Polyak-averaged targets, and automatic entropy tuning. It is well-suited to the residual VLA setting: replay reuse amortizes the expensive frozen-prior computation; automatic entropy tuning eliminates a brittle exploration schedule; and deterministic mean-action evaluation recovers precision at deployment time. PPO [10] is a natural baseline but cannot escape on-policy data discard in a single-environment regime with an expensive base policy query per step.

## III. METHOD

### A. System Overview

The system contains a frozen VLA, a trainable squashed-Gaussian residual actor, twin SAC critics, and an external reward function. Fig. 1 separates deployment-time action production from training-only replay, reward, and privileged critic inputs.

### B. Frozen VLA Base Policy and Cached-Prior Replay

We use the roughly 1B-parameter Prismatic-compatible MiniVLA checkpoint [2], whose DINO+SigLIP visual backbone is fused with a Qwen2.5-0.5B language model. Following OpenVLA [6] and Prismatic VLMs [5], image embeddings and language tokens condition autoregressive action tokens, which are decoded and unnormalized into a 7-dimensional end-effector delta and gripper action. A `MiniVLAAdapter` exposes `predict_action(image, instruction)`:

$$a_{\text{vla}} = \pi_{\text{vla}}(o_t, \ell) \tag{1}$$

where $o_t$ is the RGB image and $\ell$ the instruction. No gradients flow through the VLA. Because greedy decoding is deterministic, each transition stores the prior actions for both the current and next observations, $a_{\text{vla},t}$ and $a_{\text{vla},t+1}$. SAC updates therefore require no additional VLA forward passes.

### C. Residual Action Parameterization

The executed action is produced by the `MiniVLAResidualController`:

$$a_{\text{exec}} = \text{clip}(a_{\text{vla}} + \rho \odot \tanh(u),\ a_{\text{low}},\ a_{\text{high}}) \tag{2}$$

where $u \in \mathbb{R}^d$ is the pre-tanh actor sample and $\rho$ bounds the correction component-wise. We use $\rho_i = 0.1$ on the six end-effector dimensions and $\rho_7 = 0.5$ on the gripper; the latter is capped below an earlier, unstable setting but is larger than the end-effector bound. The outer clip enforces normalized action limits $[-1, 1]$. The mean head is zero-initialized, so deterministic mean-action evaluation before learning gives $a_{\text{exec}} = a_{\text{vla}}$; stochastic exploration need not be a no-op.

The controller logs a `ControllerOutput` dataclass at each step with $a_{\text{vla}}$, $u$, $\tanh(u)$, $a_{\text{exec}}$, and diagnostic norms (`residual_norm`, `prior_deviation_norm`, `residual_base_ratio`), which are surfaced in training logs.

### D. Asymmetric Actor-Critic Architecture

A key design choice is the information asymmetry between the actor and critic. The **actor** observes only *deployable* signals:

$$s_a = [\,\text{eef\_pos},\ \text{eef\_axis\_angle},\ \text{gripper\_qpos},\ a_{\text{vla}}\,] \in \mathbb{R}^{n_a} \tag{3}$$

Appending the cached $a_{\text{vla}}$ to the proprioceptive state gives the actor direct access to the base policy's intent without any VLA gradient flow. The critic additionally observes a fixed 13-dimensional privileged vector available only during simulation training:

$$p = [\,\text{obj\_pos},\ \vec{r}_{\text{eef}\rightarrow\text{obj}},\ d_{\text{obj}},\ \Delta z_{\text{obj}},\ q_{\text{gripper}},\ c_{\text{seen}},\ c_{\text{closed}},\ c_{\text{lift}}\,] \in \mathbb{R}^{13} \tag{4}$$

where $\vec{r}_{\text{eef}\rightarrow\text{obj}} \in \mathbb{R}^3$ is the end-effector-to-object vector, $d_{\text{obj}}$ is scalar distance, $\Delta z_{\text{obj}}$ is change in object height, $q_{\text{gripper}} \in \mathbb{R}^2$ are gripper joint positions, and $c_{\text{seen}}, c_{\text{closed}}, c_{\text{lift}} \in \{0, 1\}$ are binary flags for contact, gripper closure, and positive lift after contact. The critic observation is $s_c = [s_a; p] \in \mathbb{R}^{n_a+13}$. This vector is never passed to the actor, making the actor deployable on hardware where these signals are unavailable.

The actor $\pi_\theta$ outputs a state-dependent mean $\mu_\theta(s_a)$ and log-standard-deviation $\log \sigma_\theta(s_a)$, implementing a squashed-Gaussian policy with the tanh-corrected log-probability:

$$\log \pi_\theta(u \mid s_a) = \log \mathcal{N}(u; \mu_\theta, \sigma_\theta^2) - \sum_i \log(1 - \tanh^2(u_i)) \tag{5}$$

The Jacobian correction in Eq. (5) is necessary for unbiased policy gradients when actions pass through $\tanh$; omitting it inflates entropy estimates and biases the actor loss. The mean head is zero-initialized so the policy is a no-op in deterministic-eval mode at step 0. Log-standard-deviation is

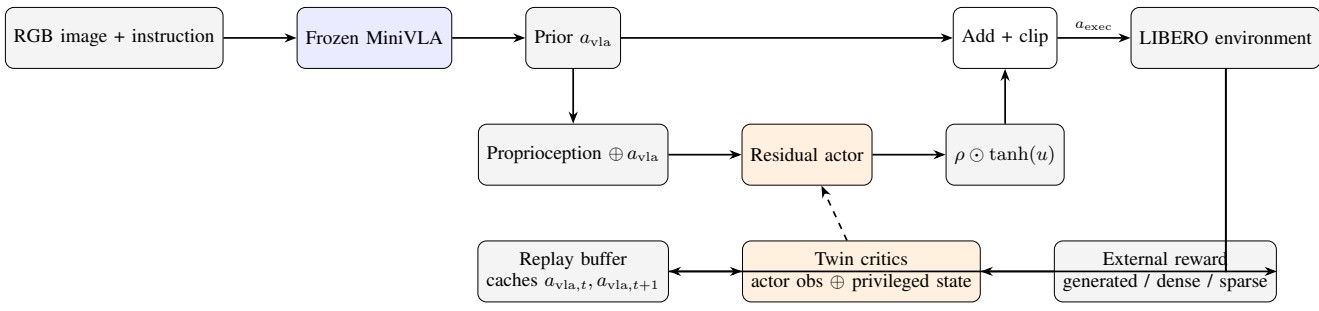

Fig. 1. R-VLA architecture. The frozen VLA produces a prior action from the image and instruction. The actor consumes only proprioception and that prior, and its bounded correction is added before environment execution. Replay, external rewards, privileged state, and critics are training-only; dashed arrows denote actor optimization through the critics.

clamped to $[\log \sigma_{\min}, \log \sigma_{\max}]$ with $\sigma_{\max} = e^{0.5} \approx 1.65$, keeping exploration within the residual correction regime.

### E. SAC Objective

Training uses SAC [3] with twin Q-critics $Q_{\phi_1}, Q_{\phi_2}$ and Polyak-averaged target critics. The Bellman backup is:

$$y_t = r_t + \gamma(1 - d_t)\Big[\min_i Q_{\phi_i^-}(s_{c,t+1}, a') - \alpha \log \pi_\theta(a' \mid s_{a,t+1})\Big] \tag{6}$$

where $a' \sim \pi_\theta(\cdot \mid s_{a,t+1})$, $\alpha$ is the automatically tuned temperature, and $d_t$ is a true-terminal flag. Time-limit truncations are bootstrapped rather than treated as terminal. The critic loss minimizes $\mathrm{MSE}(Q_{\phi_i}, y_t)$. The actor loss is:

$$\mathcal{L}_\pi = \mathbb{E}_{s_a \sim \mathcal{B}}\Big[\alpha \log \pi_\theta(a \mid s_a) - \min_i Q_{\phi_i}(s_c, a)\Big] \tag{7}$$

Temperature $\alpha$ is learned by minimizing $\mathcal{L}_\alpha = -\log \alpha \cdot (\log \pi_\theta + \mathcal{H}_{\text{target}})^\top \mathbf{1}$ with target entropy $\mathcal{H}_{\text{target}} = -d$ where $d$ is the action dimension. The scalar reward entering Eq. (6) is scaled by a factor $c_r$ before backup to balance the task signal against the entropy term $\alpha \log \pi_\theta$. During a configurable warmup phase the actor is bypassed and residual deltas are sampled from a scaled Gaussian ($0.3 \cdot \mathcal{N}(0, I)$ in pre-tanh space) to populate the replay buffer with near-prior data before gradient updates begin; full-magnitude unit-Gaussian warmup is avoided because it saturates $\tanh$ and fills the buffer with near-random corrections far from the VLA prior manifold. An earlier configuration of the reward scale $c_r$ produced a maximum-entropy collapse, in which the entropy bonus $\alpha \log \pi_\theta$ dominated the sparse task reward and the critic's implied $Q$-values diverged to $\sim +140$ with no corresponding task progress; reducing $c_r$ and re-balancing against the per-step task reward resolved this and is reflected in the well-calibrated $Q$-values reported in Section IV.

### F. GRACE-Style LLM Reward Generation

Rewards are generated offline following the GRACE methodology [9], which produces interpretable code rather than a learned black-box reward model. The prompt supplies the task description, observation and action schemas, privileged simulator fields, and a fixed function contract:

```
def compute_reward(
    obs, action, next_obs, info
) -> Tuple[float, Dict[str, float]]:
    ...
```

The `ExternalReward` wrapper loads this function, sends its scalar output to replay, and logs the diagnostic dictionary independently. Thus generated, hand-written, and sparse rewards share the same training code. The generated t9 reward audited below includes distance progress, contact and gripper-state terms, object-height and lift terms, penalties for failed grasp patterns, and a $+2$ environment-success bonus. These privileged quantities affect training rewards but are not actor inputs. We use one generation round; iterative Eureka-style refinement [8] is future work.

### G. Chunked Action Support

The controller supports an `action_horizon` parameter $H > 1$, in which the actor outputs $H$ consecutive correction vectors rather than a single-step correction, matching the chunked prediction interface of modern VLAs. In chunked mode, $a_{\text{exec}}$ becomes a matrix $\in \mathbb{R}^{H \times d}$ and Eq. (2) is applied element-wise across the horizon.

## IV. QUANTITATIVE EVALUATION ON LIBERO

### A. Protocol and Two Corrections

All experiments use MiniVLA (`minivla-libero90-prismatic`, float16, frozen) on LIBERO [7] tasks. Base-policy (E0) evaluations run 20 episodes per task, episode $i$ starting from LIBERO init state $i$; residual training cycles through all 50 init states, with periodic deterministic-actor evaluation (mean action, 10 episodes over init states 0–9). Success is read only from the environment's task-completion predicate, never from episode termination or truncation. Every reported number is generated by a logged run with seed, configuration snapshot, and git revision stored alongside its result file.

We report two protocol bugs because both invalidate numbers in our earlier workshop submission. First, that evaluation reset every episode to the same init state, so a deterministic policy replayed one trajectory $N$ times. Second, the environment adapter omitted the gripper-command sign inversion required by the reference Prismatic-family LIBERO evaluation,

causing close commands to execute as open. All results below vary init states and use the corrected adapter.

## B. Frozen-Policy Anchor (E0)

We screen the frozen MiniVLA policy on 16 LIBERO tasks before residual training. Within `libero_90`, success spans 20–100% across ten tasks; six out-of-suite `libero_spatial` and `libero_goal` tasks are 0/20 when evaluated with the available `libero_90` action-unnormalization statistics. Table I reports the three tasks carried into training: t3 and t7 represent partial base competence, while t9 is a high-base-rate stress case for preserving prior behavior.

TABLE I

E0 FROZEN-POLICY ANCHOR FOR THE THREE TASKS CARRIED INTO RESIDUAL TRAINING (OF 16 TASKS EVALUATED IN TOTAL; THE REMAINING 13 SPAN 20–100% IN-SUITE AND 0% OUT-OF-SUITE). EPISODE $i$ STARTS FROM LIBERO INIT STATE $i$.

| Task | Role | Success (20 eps) |
|------|------|------------------|
| `libero_90` t3 | partial competence | 50% |
| `libero_90` t7 | partial competence | 65% |
| `libero_90` t9 | preservation stress | 80% |

## C. Residual SAC Training Against the Anchor (E1)

We train residual SAC under generated dense rewards on the three tasks, using 3 seeds and 10,000 environment steps per run. The frozen backbone runs once per environment step at $\sim$0.35 s per forward on the Apple-silicon workstation used throughout. Table II compares deterministic evaluations at 5k and 10k steps with the separately estimated E0 anchors. The denominators differ: E0 uses 20 episodes over init states 0–19, whereas each trained checkpoint uses 10 episodes over states 0–9. Accordingly, the anchor is a reference estimate rather than a paired per-state difference.

TABLE II

E1: RESIDUAL-SAC SUCCESS UNDER THE GRACE-GENERATED REWARD (10 DETERMINISTIC EPISODES, INIT STATES 0–9; MEAN [MIN, MAX] OVER 3 SEEDS). BASE = FROZEN-POLICY ANCHOR (TABLE I, 20 EPISODES).

| Task | Base | 5k steps | 10k steps |
|------|------|----------|-----------|
| t3 | 0.50 | 0.00 [0.00, 0.00] | 0.13 [0.00, 0.30] |
| t7 | 0.65 | 0.13 [0.00, 0.20] | 0.03 [0.00, 0.10] |
| t9 | 0.80 | 0.07 [0.00, 0.10] | 0.30 [0.10, 0.40] |

Within these separately estimated protocols, every trained mean is below its frozen anchor. The t9 mean is 7% at 5k steps against an 80% 20-episode anchor, with partial and unstable recovery by 10k. Section IV-E tests whether the generated return is aligned with success; the short budget prevents attributing every failure solely to reward design.

## D. Ablations (E2)

All ablations run on t3 (base 50%), one seed, 10k steps—preliminary by construction; they bound effects rather than establish them. Table III summarizes; three findings stand out.

**Cached-prior replay reduces compute.** Because the frozen backbone and decoding are deterministic, caching changes wall-clock cost rather than the mathematical update. Matched short runs (250 environment steps, 51 updates, batch 64) take 91.6 s with caching and 2,292.8 s when both priors are recomputed for each sampled transition, a 25× slowdown. Extrapolating to batch 256 and 10k updates yields roughly 440 additional hours versus a $\sim$2.5-hour cached run.

**The symmetric critic degrades least under the same broken reward.** Removing the privileged critic input (Section III) is the least-degraded condition in Table III: 40% at both 5k and 10k, versus 0%/10% for the asymmetric (privileged) critic used throughout the rest of this paper. One consistent reading, which the audit below supports: the privileged critic observes exactly the object-position and contact quantities the shaped reward pays for, so it fits the exploitable value function faster. With one seed this is a hypothesis, not a conclusion, and the asymmetric critic remains motivated on sim-to-real grounds (Section III) independent of this reward-hacking interaction.

**Neither dense reward survives training; the sparse reward degrades differently.** The hand-written staged reward performs no better than the GRACE-generated one (0% at both checkpoints). The sparse task-success reward decays more slowly (20% at 5k) but also collapses by 10k (0%)—through a different mechanism, however: rolling *stochastic* training success stays near base (30–40%) while the deterministic-eval mean drifts (mean residual norm 0.45, episodes hitting the step cap), consistent with an unconstrained actor mean under a mostly flat $Q$-landscape rather than reward exploitation.

TABLE III

E2 ABLATIONS ON T3 (BASE 50%): DETERMINISTIC SUCCESS AFTER 5K / 10K STEPS. ONE SEED EACH.

| Condition | 5k | 10k |
|-----------|-----|-----|
| Full method (GRACE reward, asym. critic) | 0.00 | 0.10 |
| No VLA prior (zero prior) | 0.00 | 0.00 |
| Symmetric critic | 0.40 | 0.40 |
| Hand-written staged reward | 0.00 | 0.00 |
| Sparse success reward | 0.20 | 0.00 |
| Frozen base (no training) | 0.50 | |

Removing the VLA prior entirely (residual-only control, same bounds, zero prior) never succeeds: the bounded residual cannot reach the workspace unaided. The prior is what makes a small, bounded residual viable at all, independent of what happens to the reward.

## E. Reward-Hacking Audit (E3)

We audit whether generated return tracks task success using three diagnostics on t9:

**(1) High return coexists with low success.** After 5k steps, one deterministic t9 policy earns mean return 446 while succeeding in 10% of episodes and averaging ≈382 steps. Success terminates the episode, whereas shaping continues to

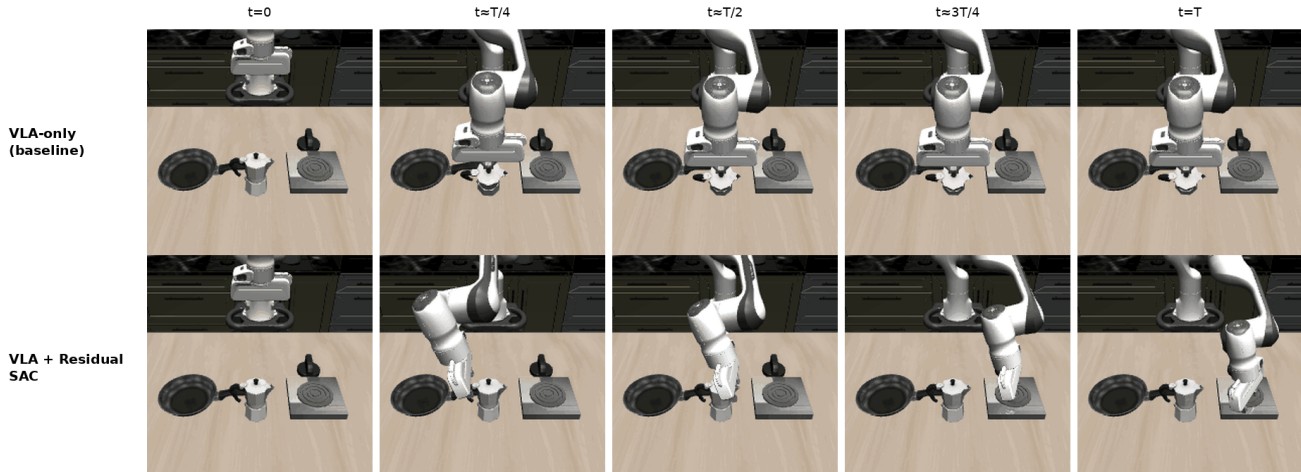

| t=0 | t≈T/4 | t≈T/2 | t≈3T/4 | t=T |

Fig. 2. Matched OOD moka-pot rollouts at five normalized timesteps. **Top:** the frozen VLA approaches the object and then stalls in nearly the same pose. **Bottom:** VLA plus residual SAC continues to intervene, descending near the moka pot and moving toward the stove rather than remaining stationary. The comparison demonstrates a directed qualitative change in behavior, not task completion.

accumulate; the generated $+2$ success bonus is small relative to the return available from prolonged interaction.

**(2) Shaped return decorrelates from success.** Pooling all E1 evaluation episodes per seed, the Pearson correlation between episode return and task success ranges from $-0.21$ to $0.66$ and is near zero for the highest-return runs; mean evaluation return varies $3\times$ across t9 seeds (179 to 554) while success stays within 10–25%.

**(3) Hack counters localize the exploit.** The highest-return t9 seed accumulates 498 contact-with-positive-object-height steps across 20 evaluation episodes—reward farmed through sustained contact and lift shaping—with zero empty-gripper cycles, i.e. the exploit is loitering near the object rather than gripper-spamming.

Under the sparse reward, return is the success indicator and its correlation with success is $1.00$. Generated-reward correlations span $-0.21$ to $0.66$. The hand-written dense condition has no successful evaluation episodes, so its correlation is undefined and cannot support the same comparison. These diagnostics flag possible reward misalignment, but do not by themselves prove that reward design is the only cause of degradation.

### F. Qualitative OOD Pilot

We also retain a single-seed pilot on the custom OOD instruction *place the moka pot on top of the frying pan*. It is not included in the quantitative comparison above and does not establish task success; instead, it illustrates how the learned residual changes a frozen policy's trajectory. In the matched rollout pair in Fig. 2, the VLA-only controller approaches the moka pot and then remains nearly stationary, whereas the residual-corrected controller continues moving through the workspace, first descending near the pot and then sweeping laterally toward the stove. The residual norm becomes nonzero after training and the corrected trajectory remains active throughout the episode. This is qualitative

progress from stalling to a directed, task-relevant intervention, although neither rollout completes the placement.

### V. DISCUSSION AND FUTURE WORK

*a) Prior-Deviation Regularization and Demo Buffer.:* In the current SAC formulation the actor is anchored to the VLA prior only through zero-initialization and the bounded `tanh` parameterization. A more explicit behavioral cloning-to-prior penalty on the actor loss—penalizing $\|\mu_\theta(s_a)\|$ during early training when the Q-surface is flat—is a natural extension. A `DemoBuffer` interface supporting RLPD-style [1] mixing of demonstration transitions into each SAC minibatch is already implemented; the `demo_sample_fraction` hyperparameter controls the mixture ratio. Evaluating this on LIBERO tasks with base-policy rollouts as demonstrations is a near-term experiment.

*b) Sim-to-Real Transfer.:* The asymmetric critic design is specifically motivated by sim-to-real transfer: because the actor observes only deployable signals, it can be deployed on hardware without modification after training, discarding the privileged critic inputs entirely. Adding dynamics domain randomization and validating the actor on a physical 7-DOF arm is a near-term priority.

*c) Reward Hacking Mitigation.:* LLM-generated rewards may contain exploitable loopholes [8]. Section IV-E identifies a concrete warning sign: return can increase through prolonged contact while success remains low. Bounded residual actions limit deviation from the prior but do not repair the incentive. Future reward candidates should be screened on known successful and unsuccessful trajectories, rejected when return is poorly aligned with success, and designed so accumulated shaping cannot dominate the $+2$ terminal bonus.

### VI. LIMITATIONS

Everything reported here is simulation-only, on LIBERO, with one frozen backbone; deployability is architectural rather

than validated on a physical robot. Training runs are short (10k environment steps), so non-monotonic curves cannot distinguish failure from insufficient optimization. E1 uses three seeds, but component ablations use one seed and one task. The frozen anchors use 20 episodes while trained checkpoints use 10, preventing paired per-state effect estimates. Reward generation is tested through reward conditions and diagnostics, not as a generation process. We do not compare against PPO, LoRA, full fine-tuning, or a stronger prior-regularized residual method; the frozen VLA is the primary comparator, and Table III is a component study rather than a competitive benchmark. Finally, the protocol corrections in Section IV-A make earlier workshop numbers incomparable to those reported here.

## VII. Conclusion

We presented R-VLA, a residual SAC architecture that keeps the VLA frozen, separates deployable actor observations from privileged critic inputs, and caches current and next prior actions in replay. Cached replay reduces measured wall-clock cost by $25\times$ relative to recomputation, and the bounded actor reproduces the frozen policy under deterministic evaluation at initialization.

After two protocol corrections, trained policies remain below separately estimated frozen-policy anchors within the tested 10k-step budget. The reward audit finds high generated return with low success, weak return–success correlation, long episodes, and sustained object contact; these are preliminary indicators of reward misalignment rather than a successful correction result or a complete causal account. The OOD pilot nevertheless shows a qualitative change from a stalled frozen rollout to an active, directed residual intervention. The contribution is therefore the residual architecture, anchor-first evaluation protocol, and audit instrumentation. Longer runs, stronger baselines, matched evaluation sets, and reward screening are required before claiming effective VLA adaptation.

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
