# OpenReview forum: "R-VLA: Residual Off-Policy Adaptation of a Frozen Vision-Language-Action Policy"
_roboticsfoundation.org/RSS/2026/Workshop/RL4VLA — RL4VLA_

### Official Review · Reviewer_fcqE · 2026-06-23
**Review of R-VLA**

**Rating:** 4
**Confidence:** 4

**Review:**

Overall, this work is highly relevant but is lacking writing quality as well as concise, motivated design choices to communicate insights. It does not come off as particularly clear,  but it does seem like it could be an original system, however I’m not confident in knowing whether others have made similar systems. Using GRACE for LLM-based rewards is fascinating and is a significant piece towards the puzzle of self-improving policies that practice in simulation. In essence, there's mostly a lack of quality and clarity but strong relevance, originality, and significance.

Comments:
- The abstract is not written very clearly, as it doesn’t provide an overarching picture that motivates the system's decisions. Instead, it reads more like a statement mentioning catastrophic forgetting (which tends to be more of a concern in continuous learning settings, not the single-task fine-tuning regime) followed by a summary of how the method works, with unorganized technical details. Once the paper is more complete, it’d be good to make the abstract more holistic and improve the flow of the writing.
- The idea of doing residual corrections on cached VLA actions is interesting, although in the introduction, it’s not entirely clear to me why the GRACE methodology is being used to auto-generate Eureka-style rewards. Maybe in the introduction, add an example of a setting where we’d need a compute-cheap training procedure (motivating the residual policy) that is sample efficient (motivating off-policy). Since it’s sim2real, part of me thinks the problem setting is a bit convoluted since simulation usually implies access to lots of offline compute. I can see how doing real-world online RL might necessitate off-policy and low latency though.
- I feel like you should find a solid problem setting where your contributions make sense. There are mentions of wanting low memory usage and no backward passes through backbones for on-edge devices. Further, there’s a mention of not wanting to use expert demonstrations either. Maybe it’d be good to solidify the setting here to make it clearer. There are mentions of sim2real, on-edge, off-policy, etc., and it is becoming unclear what motivates what - it seems you’re in the online off-policy Sim2Real reward-free setting.  If we have made the task in simulation, I am confused why we wouldn’t have reward access.
- Most of the novelty seems to lie in the composition of asymmetric sim2real SAC for doing residual learning. Maybe it’d be good to focus here. Further, I think with papers which focus on the creation of a system, there should ideally be experimental evidence to show why each component was used and what design decisions practitioners should tend to make. Or demonstration of very strong results to inform readers that today’s components are all we need to enable some tasks. I figure this is planned for later, as mentioned in the Conclusion.
- The results seem to only show a sanity check and an inconclusive negative result, so it’s a bit difficult to examine the work quality due to this incompleteness. A system figure is also required. Finding better experiments will be good too, ones which really expose what the current baselines fail to deliver in your selected problem setting.
- I appreciate the details but ideally these details would be combined with concrete insights or takeaways and ablation studies. The Discussion section gets at this, but doesn't come off as thorough enough.
- $\texttt{ExternalReward}$ is mentioned a bit but it's unclear what advances were made in the creation of this interface and why it's relevant to the reader. The same applies to Python file paths shared in the paper.

In conclusion, the system looks interesting. However, the paper in its current state is perhaps a bit too incomplete for the workshop, does not tend to provide insight towards algorithmic choices or surprising decision decisions, and lacks grounding to motivate realistic robotic settings. There may be some work required to solidify the method and sharing anything to the community. Additionally, some improvement in writing clarity would be appreciated as most writing in the results section is sharing low-level details with an unclear takeaway.

---

### Official Review · Reviewer_PJMX · 2026-06-25
**Relevant early-stage residual RL system for frozen VLAs, but quantitative evidence is still preliminary**

**Rating:** 6
**Confidence:** 4

**Review:**

This paper proposes a residual RL framework for adapting a frozen VLA. The base VLA remains fixed, and a small residual policy is trained with SAC to add a bounded correction to the VLA action. The design includes zero initialisation, cached VLA actions in replay, an asymmetric actor-critic with privileged simulator state only for the critic, and GRACE-style LLM-generated reward functions plugged in through an external reward interface. The reported experiment is a short single-seed LIBERO OOD task run. The paper does not achieve task success, but it shows that the residual policy begins to deviate from the frozen VLA and produces some intermediate reward signals.

This paper is clearly relevant to RL4VLA. It directly addresses RL-based adaptation of VLA policies, reward generation for embodied agents, off-policy training, and possible sim-to-real deployment through a deployable actor / privileged critic split. Among the two submissions, this one is much closer to the workshop’s core topic.

The system design is reasonable. Freezing the VLA and learning a bounded residual is a practical compromise between full fine-tuning and doing nothing. Cached-prior replay is also a good engineering idea: if the frozen VLA action is deterministic, storing it once per environment step can avoid repeated expensive VLA calls during SAC updates. The asymmetric critic is standard but appropriate for sim training, and the zero-initialised residual makes the “do no harm at step 0” claim fairly clean.

The main weakness is the empirical stage. The paper reports 0/5 success at every evaluation checkpoint on one task and one seed. The qualitative figure suggests that the residual policy changes behaviour, but it does not show that the method solves the task or improves success over the frozen VLA. The Q-value and reward-component curves are useful sanity checks, but they are not enough to support a strong performance claim. The reward-generation part also needs more validation: generated dense rewards can be exploited, and the current paper does not yet show reward hacking checks, human-written reward comparisons, or ablations.

Pros:
Strong match to the workshop topic: RL adaptation of VLA policies with generated rewards.
The residual formulation is practical and avoids full VLA backpropagation.
Zero initialisation, bounded residuals, cached-prior replay, and asymmetric critic are sensible design choices.

Cons:
No task success is achieved in the reported experiment.
Evaluation is only one OOD LIBERO task, one seed, and a short training horizon.
There are no strong baselines or ablations, such as SAC from scratch, residual without cached prior, residual without privileged critic, human-designed reward, or frozen VLA only over more episodes.
The GRACE-generated reward is described in detail, but its correctness and robustness are not yet validated.
The sim-to-real motivation is plausible but not tested.

---

### Decision · Program_Chairs · 2026-07-03

**Decision:**

Accept

**Comment:**

This paper presents an early-stage systems approach that freezes a vision-language-action model and trains a small bounded zero-initialized residual policy using asymmetric SAC, cached-prior replay, and GRACE-generated rewards. The reviewers agreed that the idea is relevant and well aligned with the workshop, but had mixed opinions on the current evidence. The main concerns are the limited experimental validation, including results on only one task with one seed, no strong baselines, and an inconclusive outcome, as well as unclear writing. We believe these concerns are appropriate for an early-stage systems paper and do not outweigh the value of the proposed direction. For the camera-ready version, the authors should add a system architecture figure, improve the abstract and focus on a single motivation (sim-to-real transfer, off-policy learning, or on-edge adaptation), present the current results clearly as preliminary, and clarify what inputs are provided to the residual policy.